# Characterisation of populations at risk of sub-optimal dosing of artemisinin-based combination therapy in Africa

Abena Takyi[1,2,3,4☯], Verena I. Carrara[1,2,3,5☯], Prabin Dahal[1,2,3], Marianna Przybylska[6], Eli Harriss[7], Genevieve Insaidoo[8], Karen I. Barnes[2,3,9], Philippe J. Guerin[1,2,3]*, Kasia Stepniewska[1,2,3]*

1 Centre for Tropical Medicine and Global Health, Nuffield Department of Medicine, University of Oxford, Oxford, United Kingdom, 2 Infectious Diseases Data Observatory (IDDO), Oxford, United Kingdom, 3 WorldWide Antimalarial Resistance Network (WWARN), Oxford, United Kingdom, 4 Department of Child Health, Korle Bu Teaching Hospital, Accra, Ghana, 5 Institute of Global Health, Faculty of Medicine, University of Geneva, Geneva, Switzerland, 6 Royal Infirmary Edinburgh, NHS Lothian, Edinburgh, United Kingdom, 7 The Knowledge Centre, Bodleian Health Care Libraries, University of Oxford, Oxford, United Kingdom, 8 Holy Family Hospital, Nkawkaw, Eastern Region, Ghana, 9 Division of Clinical Pharmacology, Department of Medicine, University of Cape Town, Cape Town, South Africa

☯ These authors contributed equally to this work.
* kasia.stepniewska@wwarn.org (KS); philippe.guerin@wwarn.org (PJG)

**Data Availability Statement:** Data from cited openly available sources were accessed and data

## Abstract

Selection of resistant malaria strains occurs when parasites are exposed to inadequate anti-malarial drug concentrations. The proportion of uncomplicated *falciparum* malaria patients at risk of being sub-optimally dosed with the current World Health Organization (WHO) recommended artemisinin-based combination therapies (ACTs) is unknown. This study aims to estimate this proportion and the excess number of treatment failures (recrudescences) associated with sub-optimal dosing in Sub-Saharan Africa. Sub-populations at risk of sub-optimal dosing include wasted children <5 years of age, patients with hyperparasitaemia, pregnant women, people living with HIV, and overweight adults. Country-level data on population structure were extracted from openly accessible data sources. Pooled adjusted Hazard Ratios for PCR-confirmed recrudescence were estimated for each risk group from published meta-analyses using fixed-effect meta-analysis. In 2020, of the estimated 153.1 million uncomplicated *P. falciparum* malaria patients in Africa, the largest risk groups were the hyperparasitaemic patients (13.2 million, 8.6% of uncomplicated malaria cases) and overweight adults (10.3 million, 6.7% of uncomplicated cases). The estimated excess total number of treatment failures ranged from 0.338 million for a 98% baseline ACT efficacy to 1.352 million for a 92% baseline ACT efficacy. Our study shows that an estimated nearly 1 in 4 people with uncomplicated confirmed *P. falciparum* malaria in Africa are at risk of receiving a sub-optimal antimalarial drug dosing. This increases the risk of antimalarial drug resistance and poses a serious threat to malaria control and elimination efforts. Changes in antimalarial dosing or treatment duration of current antimalarials may be needed and new antimalarials development should ensure sufficient drug concentration levels in these sub-populations that carry a high malaria burden.

used for the analysis are available in the supplementary materials.

**Funding:** Bill & Melinda Gates Foundation (grant INV-004713) and the University of Oxford. The funders of the study had no role in the study design, evidence synthesis, writing of the manuscript or the decision to submit it for publication.

**Competing interests:** The authors have declared that no competing interests exist.

## Introduction

In 2021, the World Health Organization (WHO) Africa region alone accounted for approximately 234 of the estimated 247 million malaria cases and 96% of the estimated 619,000 malaria deaths worldwide. Four sub-Saharan Africa countries contributed to about half of the total burden of cases. Increasing investment in malaria control and the scaling up of artemisinin-based combination treatment (ACT) deployment led to a steady decline of 27% in the incidence of malaria cases between 2000 and 2015. Since then, the number of malaria cases is rising again, most of the increase occurring in the African region [1]. The COVID-19 pandemic seriously disrupted healthcare systems and alongside the direct impact on malaria control programmes, in most endemic countries, access to health care remains challenging for many patients; the 2022 WHO World Malaria Report estimates that an additional 13.4 million cases and 63,000 deaths worldwide were due to disruptions during the pandemic [1].

*Plasmodium falciparum* (*Pf*) is responsible for most cases of severe malaria and the majority of malaria deaths. The continuous reduction in malaria deaths prior to the pandemic, 37% since 2000, persisted despite the increasing number of cases observed. This success might be attributed to the widespread availability of intravenous artesunate followed by an ACT for the treatment of severe malaria [2, 3].

Following the emergence and spread of *Pf* resistant strains to sequential monotherapies, namely chloroquine in the 1960s, followed by sulfadoxine-pyrimethamine in the 1980s [4, 5], and then mefloquine in the 1990s [6], the ACTs became the WHO recommended first-line treatment for uncomplicated *Pf* malaria in 2006 [7]. Since its introduction, artemisinin resistance has been reported in 2007 in Southeast Asia and in Eastern India [8–10]. Resistance to the partner drugs associated with the artemisinin derivatives is of high concern in these regions, leaving very few therapeutic options [11, 12]. With the recent confirmation of independent foci of clinically significant artemisinin resistance emerging on the African continent, specifically in Uganda, Rwanda and Eritrea, and low PCR-adjusted efficacy including in Burkina Faso and Angola, artemisinin and/or partner drug resistance could threaten malaria control and elimination efforts across the continent [13].

Resistance can arise as a consequence of spontaneous changes in the genetic structure of the parasite which provides a competitive advantage allowing it to survive the treatment even when the patient receives recommended doses of ACTs [5]. Another scenario conducive for the selection of resistant parasite strains is inadequate drug exposure [14] or sub-optimal-dosing, a situation where parasites are exposed to an insufficient drug concentration and/or for an inadequate duration to clear the infection [5]. Low drug exposure can occur for various reasons including sub-optimal recommended dose from the registered product, poor patient adherence, poor-quality medicines (either sub-standard or falsified medicines with reduced active ingredients), or inadequate absorption (e.g. acute vomiting shortly after drug administration) [15, 16]. These contributory factors may be avoidable. Absorption, distribution or metabolism of the drug, can also differ among specific groups of patients so that taking the same recommended dose in mg/kg body weight can lead to differing drug exposure [14]. Antimalarial drug pharmacokinetics are usually altered during pregnancy, due to an expanded apparent volume of distribution, resulting in lower drug levels for any given dose. Another scenario resulting in underdosing is when the standard duration of dosing is not sufficient, which might be the case in treatment of uncomplicated *Pf* malaria patients presenting with hyperparasitaemia, usually defined as parasitaemia >100,000 per microlitre. The expected treatment effect is achieved with each dose, however due to the large parasite load and the first order parasite killing process, there are still parasite left in the body when the concentration of the drug drops below the minimum inhibitory concentration (see Fig 6 in [17]).

As control efforts in Africa result in reduced transmission and case burden of infection, acquired immunity is waning, increasing the risk of more severe forms of the disease as well as resistant strains emerging and surviving in non-immune patients [18]. In the absence of alternatives to artemisinin based antimalarials in the near future, protecting the efficacy of available ACTs by identifying patient groups at high risk of receiving inadequate dosing and finding ways to optimise their treatment is paramount for the success of disease control and elimination.

The current WHO guidelines for malaria [14] identify five groups of population at risk of sub-optimal dosing: (i) malnourished children <5 years of age, (ii) pregnant women, (iii) overweight adults, (iv) patients with uncomplicated hyperparasitaemia, (v) patients co-infected with HIV or TB. WHO states that for these groups "data on antimalarial drug efficacy are still limited and insufficient evidence exists to warrant dose modification". Close monitoring of these sub-groups is strongly recommended as the risk for treatment failure and/or development of severe malaria with standard drug dosing is increased. However, the current WHO protocol for "methods for surveillance of antimalarial drug efficacy" recommends excluding severely malnourished children, cases of uncomplicated hyperparasitaemia, pregnant women and people living with HIV (PLHIV) from Therapeutic Efficacy Studies (TES) [19, 20]. Consequently, current ACT dosage regimens optimised from trials conducted initially in healthy adults and well-nourished children, must be extrapolated to these excluded populations [21].

This study aims to estimate the proportion of uncomplicated *Pf* malaria cases in endemic African countries treated with oral ACTs, as per current WHO guidelines, who are at risk of receiving sub-optimal dosing, and to estimate the fraction of treated patients likely to fail treatment because of sub-optimal dosing.

## Methods

African countries with a malaria transmission intensity estimated at one or more cases per 1000 population in 2020 [22] were included (Table A in S1 Text).

### Estimation of number/proportion of malaria cases in risk groups

Country-level data were extracted from seven public, openly accessible data sources and twenty-seven variables were retrieved which included malaria indicators, population by gender, age group, nutritional status and rural or urban areas, as well as the number of People Living with HIV (PLHIV) and fertility data Table B in S1 Text). Number of pregnant women per country is not collected routinely and was estimated using a published methodology [23] (S1 Text). Where available, lower and upper limits for the estimates were also extracted.

Number of confirmed uncomplicated malaria cases was calculated as the number of confirmed malaria cases by microscopy and/or rapid diagnostic tests (RDTs) from the WHO World Malaria Report 2021 minus the number of severe malaria cases derived from the work from Camponovo et al. who provided estimates of inpatient severe malaria cases per 100,000 persons for 41 African countries, 39 of which are part of this analysis [24]. Severe malaria cases for countries without available data were estimated using the median proportion of severe cases in countries from the same region (defined as per United Nations M49 Standards) [25].

Malaria risk was considered four times higher in rural areas than in urban settings based on published entomological inoculation rate estimates in sub-Saharan Africa countries [26–28]. Levels of malaria endemicity were categorised as hypo-endemic if *Pf* rate in children aged 2–9 years of age was ≤10%; meso-endemic if parasite rate was 11–50%; or hyperendemic if >50% [29]. The distribution of malaria cases across age categories was estimated using a model

developed by Griffin et al. [30] which defines the relationship between parasite prevalence in 2–10 years old and the proportion of malaria cases in <5, 5–9, 10–15, and >15 years age groups.

We assumed the proportion of malaria cases among acutely malnourished children <5 years of age (defined as wasted: weight-for-height <-2SD), pregnant women, PLHIV and over-weight adult population (defined as a body-mass index (BMI) > 25kg/m2) to be the same as that reported for the overall country population. However, differences in prevalence within those sub-populations between urban and rural areas have been reported in the literature [31–34] and we therefore included a risk ratio in the calculations to estimate the number at risk of malaria among each sub-population (further details in S1 Text). Hyperparasitaemia was defined as a parasite density >100,000 parasites per microliter based on two meta-analyses defining this as the threshold for an increased risk of treatment failure [35, 36]. The proportion of patients with hyperparasitaemia was assumed to be the same for each population category (as no interactions were reported in the meta-analyses) and for rural or urban settings, and based on the proportion of 10.2% (5,200/50,859, [95% CI 9.8–13.8] derived from the logistic model with random effects for study-site, personal communications) derived from an individual patient meta-analysis of over 50,000 patients from 29 African countries [37].

To ensure that malaria cases were only classified into one risk group (and therefore counted only once), sequential calculations within each age group were performed in the following order: i) children <5 years of age: wasted, living with HIV, hyperparasitaemic; ii) children 5–14 years of age: living with HIV, hyperparasitaemic; iii) adults >14 years: pregnant, living with HIV, overweight, hyperparasitaemic.

Since the objective of this study was to characterize the problem of sub-optimal dosing in patients treated with oral ACTs as per current WHO guidelines, 100% adherence and 100% ACT coverage was assumed. This represents the best-case scenario, as patients with non-adherence or not receiving recommended ACTs are at risk of sub-optimal doing, independently of their age or other risk factors.

## Estimation of failure rates on ACTs for sub-population of interest

Absolute and relative estimates of PCR-confirmed recrudescence were extracted from published meta-analyses or systematic literature reviews, searched for on Epistemonikos (S2 Text). Two additional systematic reviews were conducted to collate necessary data to support this analysis: one on the efficacy of ACTs in PLHIV (Prospero registration CRD42018089860, study ongoing), and another in non-pregnant, overweight or obese adults (Prospero registration CRD42018090521, available in S3 Text).

Where available, fixed-effect pooled estimates from meta-analyses' Hazard Ratios (HR) were calculated by risk group of interest. Otherwise, risk of treatment failure was derived from individual studies and a sensitivity analysis was performed assuming HR range 1.2–2.0. A 2–8% range of hypothetical treatment failure rates in adequately dosed patients was considered, given the current resistance data available from Africa and WHO recommendations to change drug policy if Adequate Clinical and Parasitological Response (ACPR) rate falls below 90% [19].

Calculation of the number of patients within subgroups of interest was conducted using the reported point estimates, and using lower and upper limits (where available) to provide a measure of uncertainty in our estimates. For five countries, the range values were not available for the estimated proportion of children <5 years of age who were wasted in five countries, and therefore for these countries the reported point estimate was used in the calculation of uncertainty limits. Similarly, calculations of the excess number of treatment failures were repeated

for the most/least conservative scenario, using the lowest /highest number of patients within the subpopulations together with the corresponding lower/upper limit of the 95%CI for the risk of treatment failure. For the subgroups without risk estimates, the sensitivity analysis provides a measure of their impact on the overall estimates. We refer to the lower and upper limits of our estimates as uncertainty range (UR).

### Patient and public involvement

Patients and/or the public were not involved in the design, or conduct, or reporting, or dissemination plans of this research. Since this study is a secondary analysis of publicly available data it was not possible to engage patients or public in this study.

## Results

### Number of malaria cases

Of 154.6 million confirmed cases, 153.1 million were estimated to be due to uncomplicated malaria of which 37.4 million (24.4%) were in children <5 years of age, 56.1 million (36.6%) in those 5–14 years of age, and 59.6 million (39.0%) in adults >14 years. Country-specific extracted data are provided in S4 Text, and the estimated number of malaria cases in subgroups in each country are shown in S1 Table, S1 and S2 Figs. Uncertainty limits are shown in S2 Table. Patients with hyperparasitaemia (13.2 million, UR 13.0–17.3 million, 8.6% of uncomplicated malaria cases) and overweight adults (10.3 million, UR 8.1–12.6 million, 6.7% of uncomplicated cases) were the largest risk groups in all regions and endemicity areas (Table 1). Malaria in wasted children was estimated to reach 2.5 million (UR 2.2–2.8 million), representing 1.6% of all uncomplicated cases. There were 6.4 million uncomplicated cases in pregnant women (UR 6.1–6.6 million), 4.2% of total malaria burden and 10.7% of cases in adult population. The highest proportions of PLHIV and of pregnant women at increased risk of sub-optimal dosing were in East Africa (1.5% and 2.9%, respectively), while wasted children were predominant in meso-endemic regions (2.4% *vs.* 0.1% in hypo-endemic areas), Table 1.

Distribution of estimated malaria cases across risk groups varied between countries (Fig 1 and S1 Table, S1 Fig). The proportion of PLHIV with malaria varied between <0.1 and 4.4% in all countries except Zimbabwe and Namibia, where this sub-population harboured an estimated 7.9 and 8.0% of all uncomplicated *Pf* cases respectively. The proportion of overweight adults varied between 10 and 32% of adults with uncomplicated *Pf* malaria. Proportion of wasted children among children under 5 years with uncomplicated *Pf* malaria was the highest in South Sudan (24%) and Djibouti (30%).

### Number of treatment failures

The systematic review identified five IPD meta-analyses which provided malaria recrudescence HR estimates for hyperparasitaemic patients, and one IPD for malnourished children <5 years of age (S2 Text, S4 Table, S3 Fig). Individual studies provided estimates for PLHIV (n = 4) (Table 2). No relevant studies were identified for overweight or obese patients (S3 Text) nor pregnant women.

At drug efficacy of 98%, 95% and 92%, the expected number of PCR-corrected treatment failures (malaria recrudescences) were estimated as: 3.1, 7.6 or 12.3 million, and the number of excess failures as 0.3 (UR 0.2–0.6), 0.8 (UR 0.5–1.4) or 1.4 (UR 0.9–2.2) million, respectively (assuming HR = 1.5 for pregnant women and overweight patients) (Table 3, S5 Table). The largest contribution to the excess number of treatment failures came from hyperparasitaemic patients (39.1%), (Table 3, Fig 2). Overweight adults, pregnant women, and PLHIV

**Table 1. Estimated number (in millions) of uncomplicated malaria cases per sub-population at increased risk of sub-optimal dosing.** For the uncertainty range of these estimates, see S3 Table.

| | | Wasted (in <5 years) | Pregnancy (in females >14 years) | Overweight (in >14 years) | PLHIV (in all ages) | Hyperparasitaemia (in all ages) |
|---|---|---|---|---|---|---|
| Total (41 countries) | N | 2.5 | 6.4 | 10.3 | 1.9 | 13.2 |
| | % | 1.6 | 4.2 | 6.7 | 1.2 | 8.6 |
| *By region* | | | | | | |
| Northern Africa (1 country) | N | <0.1 | 0.1 | 0.2 | <0.1 | 0.1 |
| | % | 0.0 | 0.1 | 0.1 | 0.0 | 0.1 |
| East Africa (15 countries) | N | 0.5 | 2.9 | 4.7 | 1.5 | 5.2 |
| | % | 0.3 | 1.9 | 3.1 | 1.0 | 3.4 |
| West Africa (15 countries) | N | 1.1 | 2.0 | 3.6 | 0.2 | 4.8 |
| | % | 0.7 | 1.3 | 2.4 | 0.1 | 3.1 |
| Central Africa (9 countries) | N | 0.9 | 1.4 | 1.8 | 0.2 | 3.1 |
| | % | 0.6 | 0.9 | 1.1 | 0.1 | 2.0 |
| Southern Africa (1 country) | N | <0.1 | <0.1 | <0.1 | <0.1 | <0.1 |
| | % | 0.0 | 0.0 | 0.0 | 0.0 | 0.0 |
| *By endemicity[1]* | | | | | | |
| Hypo-endemic (16 countries) | N | 0.1 | 1.0 | 2.0 | 0.3 | 1.5 |
| | % | 0.1 | 0.7 | 1.3 | 0.2 | 0.9 |
| Meso-endemic (25 countries) | N | 2.4 | 5.4 | 8.3 | 1.6 | 11.7 |
| | % | 1.5 | 3.5 | 5.4 | 1.0 | 7.7 |

Percentages are in total of estimated uncomplicated cases. The list of countries by region and by endemicity areas is found in S4 Text.

[1] Hypo-endemicity: *Plasmodium falciparum (Pf)* prevalence in 2–9 years old <10%; Meso-endemicity: *Pf* prevalence in 2–9 years old 11–50%. No country was reported as hyper-endemic in 2020.

In <5 years: children under 5 years old; in >14 years: adults aged 15 years and older.

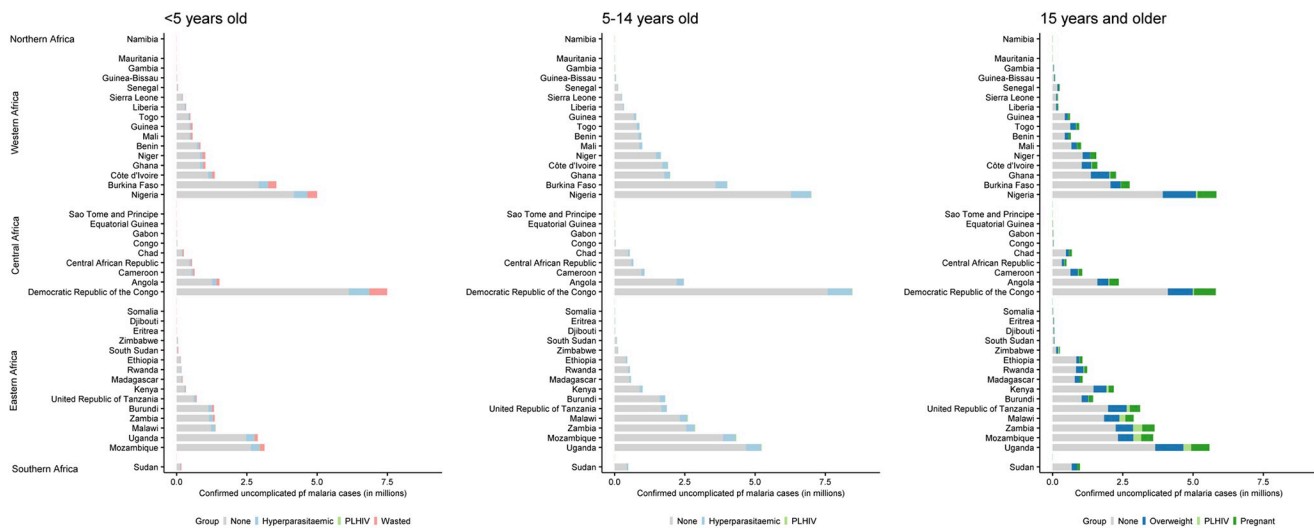

**Fig 1. Estimated number (in million) of uncomplicated *Pf* malaria cases by country and region, showing sub-population distribution with increased risk of sub-optimal dosing.**

**Table 2. Risk of treatment failure by age group and sub-population at increased risk of sub-optimal dosing used in calculation of the excess number of malaria infections[1].**

| Risk groups | Hazard Ratios [95%CI] | References to IPD meta-analyses or individual studies |
|---|---|---|
| <5 years | | |
| Hyperparasitaemic | 1.50 [1.21–1.86] (Pooled) | WWARN A-L Dose Impact SG, 2015 (PMID 25788162)<br>Saito M, 2020 (PMID 32530424)<br>WWARN DP SG, 2013 (PMID 24311989)<br>WWARN AS-AQ SG, 2015 (PMID 25888957) |
| PLHIV | 1.5 (from individual studies in Uganda and Zambia) | Kajubi R, 2016 (PMID 5170492)<br>Kamya MR 2006 (PMID 1925269)<br>Parikh S, 2016 (PMID 4946019)<br>Van Geertruyden JP, 2006 (PMID 16960779) |
| Wasted | 1.41 [1.07; 1.86] | Stepniewska K, 2016 (65th annual meeting ASTM&H, conference paper) |
| None of these | 1.0 | |
| 5 to 14 years | | |
| Hyperparasitaemic | 1.50 [1.21–1.86] (Pooled) | WWARN A-L Dose Impact SG, 2015 (PMID 25788162)<br>Saito M, 2020 (PMID 32530424)<br>WWARN DP SG, 2013 (PMID 24311989)<br>WWARN AS-AQ SG, 2015 (PMID 25888957) |
| PLHIV | 1.5 (from individual studies in Uganda and Zambia) | Kajubi R, 2016 (PMID 5170492)<br>Kamya MR 2006 (PMID 1925269)<br>Parikh S, 2016 (PMID 4946019)<br>Van Geertruyden JP, 2006 (PMID 16960779) |
| None of these | 1.0 | |
| >14 years | | |
| Hyperparasitaemic | 1.50 [1.21–1.86] (Pooled) | WWARN A-L Dose Impact SG, 2015 (PMID 25788162)<br>Saito M, 2020 (PMID 32530424)<br>WWARN DP SG, 2013 (PMID 24311989)<br>WWARN AS-AQ SG, 2015 (PMID 25888957) |
| PLHIV | 1.5 (from 2 individual studies) | Kajubi R, 2016 (PMID 5170492)<br>Kamya MR 2006 (PMID 1925269)<br>Parikh S, 2016 (PMID 4946019)<br>Van Geertruyden JP, 2006 (PMID 16960779) |
| Overweight | 1.5 (Assumed) | |
| Pregnant | 1.5 (Assumed) | |
| None of these | 1.0 | |

[1] HR for treatment failure associated with hyperparasitaemia or with HIV was assumed to be the same across all age groups.

contributed to 30.4%, 18.8%, and 5.7% of excess failures, respectively, which, in a sensitivity analysis, changed to 18.1%, 11.2%, and 3.4% (HR = 1.2 assumed) and to 39.2%, 24.3%, and 7.3% (HR = 2.0 assumed), respectively. Wasted children contributed to 6.0% excess failures. For the uncertainty limits of these estimates see S5 Table.

## Discussion

This study estimated that nearly one in four uncomplicated *Pf* malaria patients in Africa are within a sub-population of patients considered at risk of sub-optimal ACT dosing by the WHO [1]. We estimated that excess annual treatment failures could range between 0.23 to 0.52 million, 0.57 to 1.31 million and 0.91 to 2.09 million individuals in the five identified sub-populations with an ACPR of 98%, 95% and 92%, respectively.

Until optimised dosage regimens are defined for these groups, the close monitoring of treatment response in all those at risk of sub-optimal dosing will become paramount to successfully limit the emergence and spread of artemisinin- and partner drug- resistant parasite

**Table 3. Excess failures (in millions) estimations in different risk groups, assuming different treatment failure rates (FR) and a range of assumed Hazard Ratios (HR) for PLHIV, pregnant women, and overweight adults.** For the uncertainty limits of these estimates, see S5 Table.

| Risk groups | Main analysis | | | Sensitivity analyses | | | | | |
|---|---|---|---|---|---|---|---|---|---|
| | Assumed HR = 1.5 for PLHIV, Overweight and Pregnant | | | Assumed HR = 1.2 for PLHIV, Overweight and Pregnant | | | Assumed HR = 2 for PLHIV, Overweight and Pregnant | | |
| | 2% | 5% | 8% | 2% | 5% | 8% | 2% | 5% | 8% |
| | FR | FR | FR | FR | FR | FR | FR | FR | FR |
| **<5 years** | | | | | | | | | |
| Hyperparasitaemic | 0.036 | 0.089 | 0.143 | 0.036 | 0.089 | 0.143 | 0.036 | 0.089 | 0.143 |
| PLHIV | 0.001 | 0.002 | 0.003 | 0.000 | 0.001 | 0.001 | 0.002 | 0.004 | 0.007 |
| Wasted | 0.020 | 0.051 | 0.082 | 0.020 | 0.051 | 0.082 | 0.020 | 0.051 | 0.082 |
| Sub-total for <5y | 0.057 | 0.143 | 0.228 | 0.057 | 0.141 | 0.226 | 0.058 | 0.145 | 0.232 |
| **5–14 years** | | | | | | | | | |
| Hyperparasitaemic | 0.057 | 0.143 | 0.228 | 0.057 | 0.143 | 0.228 | 0.057 | 0.143 | 0.228 |
| PLHIV | 0.001 | 0.004 | 0.006 | 0.001 | 0.001 | 0.002 | 0.003 | 0.007 | 0.012 |
| Sub-total for 5-14y | 0.059 | 0.146 | 0.234 | 0.058 | 0.144 | 0.231 | 0.060 | 0.150 | 0.240 |
| **>14 years** | | | | | | | | | |
| Hyperparasitaemic | 0.039 | 0.098 | 0.157 | 0.039 | 0.098 | 0.157 | 0.039 | 0.098 | 0.157 |
| PLHIV | 0.017 | 0.042 | 0.067 | 0.007 | 0.017 | 0.027 | 0.034 | 0.084 | 0.135 |
| Overweight | 0.103 | 0.257 | 0.411 | 0.041 | 0.103 | 0.164 | 0.205 | 0.514 | 0.822 |
| Pregnant | 0.064 | 0.159 | 0.255 | 0.025 | 0.064 | 0.102 | 0.127 | 0.318 | 0.509 |
| Sub-total for >14y | 0.222 | 0.556 | 0.890 | 0.113 | 0.281 | 0.450 | 0.406 | 1.014 | 1.623 |
| **Overall TOTAL failures** | 0.338 | 0.845 | 1.352 | 0.227 | 0.567 | 0.907 | 0.524 | 1.309 | 2.094 |

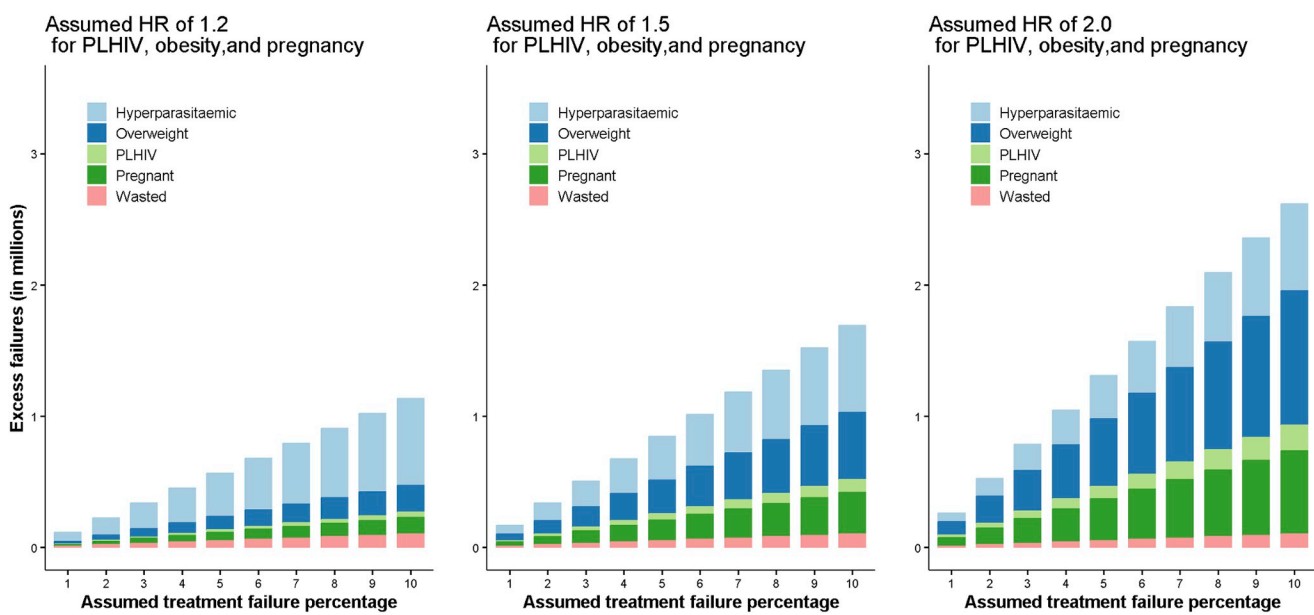

**Fig 2. Estimated number of excess treatment failures in millions for different baseline treatment efficacy assuming Hazard Ratio (HR) of 1.2, 1.5 and 2.0 in patients living with HIV (PLHIV), overweight adults, and pregnant women.**

strains on the African continent. This is especially important at a time when clinically significant artemisinin resistance has been confirmed in at least three African countries [13] and when acquired immunity is waning in regions successfully controlling the overall malaria burden [18]; and is part of the new WHO strategy to minimize the threat and impact of antimalarial drug resistance in Africa [13].

Patients with uncomplicated hyperparasitaemia accounted for 13.2 million, or 8.6%, of estimated uncomplicated malaria cases and are the largest contributor of estimated excess treatment failures. In this study uncomplicated hyperparasitaemia was defined as >100,000 parasites/μL, based on two meta-analyses defining this as the threshold for an increased risk of treatment failure [35, 36] and its proportion based on a meta-analysis on 56,000 individual patients' data that included 29 African countries in low, moderate, and high malaria transmission areas [37]. This proportion however may be an underestimation as patients with this level of parasitaemia are often excluded from uncomplicated malaria clinical trials [20]. Patients with uncomplicated hyperparasitaemia and without other clinical signs of severity are an important reservoir of de-novo resistance [17]. Additionally, inadequate treatment may aggravate the patient's clinical condition and increase risk of death [38]. Although severely ill hyperparasitaemic patients are likely to be hospitalised and treated parentally, recognising a patient with isolated uncomplicated hyperparasitaemia is challenging as diagnosis is usually made by qualitative malaria RDTs and without microscopic confirmation of parasite density. These patients are thus likely to receive a standard oral ACT dosage regimen that may be insufficient to reduce their high parasite biomass thus increasing the risk of recrudescence [17]. This problem could be rectified with use of semi-quantitative RDTs, which would allow to identify patients with parasitaemia above certain threshold. Although 100,000 parasites/microlitre have been used in the past to define hyperparasitaemia, the 'optimal' cutoffs may vary between different patient populations, different transmission intensity areas, and for different ACTs so further studies are needed to refine this definition. Currently, even if the diagnosis of uncomplicated hyperparasitaemia is made, the treatment remains problematic as evidence to date to support, e.g. increasing the ACT duration, is insufficient [14, 39]. Malaria and undernutrition often coincide in Africa, where approximately 1 in 3 children under 5 years of age are underweight (S2 Fig). The risk of malaria and treatment failure according to nutritional status remains complex [40]. Furthermore, malnutrition may worsen the severity of malaria and increase the risk of malaria deaths [41] and acutely undernourished (wasted) children are at an increased risk of ACT treatment failure [42]. This is reflected within the current WHO malaria treatment guidelines when referring to children "malnourished" as being at risk of sub-optimal dosing. In 2019, a year prior to the COVID-19 pandemic, an estimated 12.7 million children <5 years of age in Africa were acutely malnourished, of whom 3.5 million were considered severely wasted (weight-for-height Z-score <-3SD) and at higher risk of infection, complications and death [43]. The many social, economic, and health-related disruptions triggered by the COVID-19 pandemic alongside the current food shortage due to the war in Ukraine aggravate the nutritional status of an additional 1.46 million children in Africa [44]. The present study estimates that 2.5 million moderately (weight-for-height Z-score <-2SD) wasted children <5 years of age suffer from uncomplicated malaria; unless they present with danger signs or complications from their malnutrition status, these children are likely to be treated with ACTs including those in nutrition rehabilitation [45]. There is strong evidence from studies conducted in Mali and Niger that severely wasted children, treated with a full course of artemether-lumefantrine and high-fat nutritional supplements, have decreased drug exposure and a higher risk of reinfection compared to those who are well-nourished [46]. Importantly, even mild wasting (weight-for-height Z-score <-1SD) increases the risk of treatment failure to the above estimates underestimate the total effects of wasting on ACT treatment failure [42].

Furthermore, in the WWARN individual pharmacokinetic-pharmacodynamic data analysis of patients treated with artemether-lumefantrine, underweight (weight-for-age Z score $<$-2 SD) children under 3 years of age had a 23% [95%CI 1; 41] lower day 7 lumefantrine concentration [47] and underweight African children $<$3 years of age had a higher risk of treatment failure (HR 1.66 [95%CI 1.05; 2.63]) compared to adequately-nourished children of the same age [36]. Improving our understanding of the complex interactions between nutritional status, antimalarial drug absorption and ACT efficacy is paramount to improve clinical management of these patients and avoid preventable treatment failures and increasing antimalarial resistance.

In this study, persons who are overweight accounted for an estimated 10.3 million, or 6.7% of all estimated uncomplicated malaria cases, the second largest risk group. The pharmacological profile of lipophilic antimalarial drugs in overweight or obese people may be altered. One small recent pharmacokinetic study on healthy males showed non-significantly lowered artemether-lumefantrine plasma drug concentrations with higher body-weight, but was likely underpowered with only 7 overweight and 3 obese participants included [48]. Publications evaluating their risks of sub-optimal dosing [49], recrudescence or even severity are still too sparse to provide reliable estimates of effect, which is expected to vary with both the degree of obesity, antimalarial used, and level of immunity among adults enrolled. One study conducted in Sweden retrospectively reviewed medical charts of patients hospitalised with falciparum malaria and concluded that median body mass index (BMI) in patients with severe malaria was significantly higher (29.3 kg/m$^2$) than for those with uncomplicated malaria, concluding obesity (BMI $\geq$30 kg/m$^2$) was significantly associated with severe malaria at diagnosis [50]. A study by Hatz *et al.* in 165 non-immune adults reported a decreased artemether-lumefantrine day-28 parasitological cure rate (93.4% [95%CI 85.3; 97.8] in patients $\geq$65 kg compared to those $<$65 kg (100% [95%CI 92.5; 100]) [51]. In principle, dosing of ACTs should be based on a target mg/kg body weight dose, but ACTs are mostly available as pre-packaged treatments based on a single adult weight-band (e.g. artemether-lumefantrine dosage is identical for anyone weighing $\geq$35 kg) [14]. Increasing the treatment dose or prolonging the treatment regimen [52] for overweight patients could be feasible; however, it may be challenging in some primary health care contexts. As malaria transmission intensity decreases, the age distribution of malaria morbidity and mortality burden expands, with increased prevalence of malaria in the adult population. In parallel an increase in the prevalence of overweight/obesity in African adults has also been observed [53–55]. Therefore, improving diagnosis and treatment in older age groups remains relevant to advance elimination and delay resistance [30]; thus overweight adults should be actively included in dose optimization studies to provide data on this important population.

PLHIV could contribute to 1.2% of all estimated uncomplicated malaria cases and between 2.6% and 6.6% of estimated excess failures. As antiretroviral therapy (ART) coverage increases [56] together with a shift towards dolutegravir-based ARTs that have fewer drug-drug interactions [57], PLHIV may become less at risk of sub-optimal ACT dosing with standard 3-day regimen; this risk remains however for those receiving rifampicin-based tuberculosis treatment or efavirenz-based ARTs [58, 59]. Furthermore, PLHIV have higher parasites densities and children infected with HIV have been reported having slower parasite clearance than HIV-free children [60]. A recent review on the role of HIV infection on malaria transmission suggests a higher risk of re-infection in population infected with HIV-1 [61].

The current WHO Malaria guidelines for treating uncomplicated malaria in pregnancy recommend that artemether-lumefantrine should be used in all trimesters [14]. However, artemether-lumefantrine, the most widely used ACT in Africa, had a lower PCR-corrected cure rate compared to other ACTs in a large IPD meta-analysis evaluating the efficacy and

tolerability of ACTs in pregnancy [62], which could be attributed to changes in the pharmaco-kinetics of lumefantrine during pregnancy resulting in lower drug concentration compared to non-pregnant population [63]. Longer artemether-lumefantrine regimens have been tested in Thailand and in the Democratic Republic of Congo, with a higher Day-7 lumefantrine concentration compared to the standard 3-day regimen but did not show increase ACPRs [64]. Further studies to optimise antimalarial drug treatment in pregnancy are needed, as are harmonised antimalarial therapeutic efficacy assessments in pregnancy studies [65].

## Study limitations and assumptions

This study provides an estimate of the significant magnitude of the population at risk of sub-optimal dosing living in 41 African countries; those estimates are based on the latest malaria and population data openly available and are derived from several sources with some assumptions. Firstly, the number of uncomplicated malaria patients and their distribution across age structure was derived from modelling studies as only the estimated total number of malaria cases per country is available from WHO. Estimates have been calculated assuming an equal risk for everyone in a population sub-group and an equal risk by age category within that sub-group. The estimated incidence of severe malaria cases from 2015 was applied to calculate uncomplicated episodes from the 2020 total malaria data reported by WHO although malaria trends were decreasing until 2019. Recent IPD meta-analysis were not available for each population category nor were exclusively evaluating treatment failure risk in Africa. Risks of treatment failure associated with multiple factors could not be evaluated (e.g., hyperparasitaemia in pregnancy). Because sub-group populations and malaria endemicity levels were extracted at country level, granularity of risk may have been lost including level of transmission or impact of seasonality. We did not account for the quality of antimalarials (either substandard or falsified), the impact of other co-morbidities on the drug absorption, the impact of drugs other than antiretrovirals e.g. antituberculosis drugs or the true adherence to the treatment. We have attempted to provide a measure of uncertainty in the estimates of number of cases in population at risk of sub-optimal dosing and estimated the number of excess treatment failures, and have included upper and lower limits of the extracted parameter values in our calculations. However, this was not possible for all parameters. The ranges presented are therefore only an approximation of the extend of uncertainty in our predictions.

We believe that the majority of our assumptions are likely to underestimate the true overall impact of under-dosing, so provide a "best case" scenario.

## Conclusion

This study estimates that nearly 1 in 4 people with uncomplicated confirmed malaria in Africa are at risk of sub-optimal antimalarial drug dosing. This is the first attempt to quantify this issue, which poses a serious threat to malaria control efforts. Adequate antimalarial drug dosing is essential for both maximising cure rates and the prevention or delay of resistance emergence or its expansion. Optimised drug dosing or longer treatment duration of currently used ACTs may be needed in those at risk of sub-optimal antimalarial drug dosing. The largest contribution to the excess number of treatment failures came from hyperparasitaemic patients. A malaria diagnosis that includes a quantitative or semi-quantitative parasite count at all levels of health care would be of great public health value to identify patients with uncomplicated hyperparasitaemia who should receive an adapted treatment. New antimalarials should be evaluated to provide sufficient drug concentrations not only in otherwise healthy adults, but also to all at risk sub-populations that carry a high malaria burden.

## Supporting information

**S1 Text. Methods for estimation of number/proportion of malaria cases in risk groups.**
(DOCX)

**S2 Text. Systematic review to estimate failure rates of ACTs in sub-populations of interest.**
(DOCX)

**S3 Text. Systematic review of the efficacy of artemisinin-based combination therapy (ACT) in adults in Africa with uncomplicated Plasmodium falciparum malaria who are overweight or obese.**
(DOCX)

**S4 Text. Country-specific data.**
(DOCX)

**S1 Table. Estimated number of uncomplicated Pf malaria cases (in million) in population at increased risk of sub-optimal dosing, by country.**
(DOCX)

**S2 Table. Uncertainty limits of the estimated number of the uncomplicated Pf malaria cases (in million) in population at increased risk of sub-optimal dosing by country.**
(DOCX)

**S3 Table. Uncertainty limits of the estimated number of the uncomplicated Pf malaria cases (in million) in population at increased risk of sub-optimal dosing by region.**
(DOCX)

**S4 Table. Estimates of risk of malaria recrudescence in populations of interest, extracted from literature.**
(DOCX)

**S5 Table. Uncertainty limits of the estimated excess failures (in millions) in different risk groups.**
(DOCX)

**S1 Fig. Uncomplicated Pf malaria cases in populations at increased risk of sub-optimal dosing by country and region.** Data presented as percentage of total uncomplicated Pf malaria cases. Legend: pale blue = no risk factor; blue = patients with hyperparasitaemia; green = people living with HIV; red = children <5 years of age wasted; pale green = overweight adults; pale red = pregnant women.
(TIFF)

**S2 Fig. Prevalence of wasting and malaria in 41 African countries.** GeoNames, Microsoft, OpenStreetMap, TomTom. Panel A: moderate wasting (weight-for-height Z-score <-2) prevalence in children<5 years of age: <5% (pale blue), 5-<10% (blue) and ≥10% (dark blue). Latest data available from the WHO Global Health Observatory Data Repository [https://www.who.int/data/gho] are: 2020 for Rwanda, Nigeria, Gambia; 2019 for Guinea-Bissau, Ethiopia, Chad, Burundi, Liberia, Sierra Leone, Mali, Malawi, Senegal, CAR, Niger, Burkina Faso, Zimbabwe; 2018 for Guinea, Benin, Zambia, Cameroon, Tanzania, Mauritania, Madagascar; 2017 for Ghana, Togo, DRC; 2016 for Côte d'Ivoire, Uganda; 2015 for Mozambique, Angola; 2014 for Kenya, Congo, Sudan; 2013 for Namibia; 2012 for Gabon, Djibouti; 2011 for Equatorial Guinea; 2010 for South Sudan, Eritrea; 2009 for Somalia. Panel B: The 2020 Malaria Atlas Project [https://malariaatlas.org/] predicted age-standardized parasite rate for Pf malaria for

children 2–10 years of age: ≤10% (pale green), 11–25% (green) and 26–50% (dark green).
(TIFF)

**S3 Fig. Pooled estimates of the risk of treatment failure in children <5 years of age wasted, and in patients with hyperparasitaemia from all age groups.** Panel B: For DP, the Hazard Ratio (HR) was reported for Day 42; for AL and AS-AQ, the HR was reported for Day 28. The assumption of proportional hazards was satisfied for all models. Comparator groups were children <5 years of age not wasted (Panel A); patients with parasitaemia <100000 parasites/microliter (Panel B).
(TIFF)

## Acknowledgments

The authors would like to thank Jamie T. Griffin and colleagues for sharing the data used in their model for estimating the proportion of malaria cases in each age group according to the different types of malaria transmission (Fig 3c from the manuscript entitled Griffin, J.T., N.M. Ferguson, and A.C. Ghani, *Estimates of the changing age-burden of Plasmodium falciparum malaria disease in sub-Saharan Africa.* Nat Commun, 2014. **5**: p. 3136).

## Author Contributions

**Conceptualization:** Abena Takyi, Philippe J. Guerin, Kasia Stepniewska.

**Data curation:** Abena Takyi, Verena I. Carrara, Marianna Przybylska.

**Formal analysis:** Abena Takyi, Verena I. Carrara, Prabin Dahal, Kasia Stepniewska.

**Funding acquisition:** Philippe J. Guerin.

**Investigation:** Abena Takyi, Verena I. Carrara, Marianna Przybylska, Eli Harriss, Genevieve Insaidoo, Kasia Stepniewska.

**Methodology:** Abena Takyi, Verena I. Carrara, Karen I. Barnes, Philippe J. Guerin, Kasia Stepniewska.

**Supervision:** Kasia Stepniewska.

**Validation:** Prabin Dahal, Kasia Stepniewska.

**Writing – original draft:** Abena Takyi, Verena I. Carrara, Prabin Dahal, Karen I. Barnes, Kasia Stepniewska.

**Writing – review & editing:** Abena Takyi, Verena I. Carrara, Prabin Dahal, Marianna Przybylska, Eli Harriss, Genevieve Insaidoo, Karen I. Barnes, Philippe J. Guerin, Kasia Stepniewska.

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
