## [Decision Letter · Decision Letter 0]

2 Aug 2023

PGPH-D-23-00952

Characterisation of populations at risk of sub-optimal dosing of artemisinin-based combination therapy in Africa

Dear Dr. Stepniewska,

Thank you for submitting your manuscript to PLOS Global Public Health. After careful consideration, we feel that it has merit but does not fully meet PLOS Global Public Health’s publication criteria as it currently stands. Therefore, we invite you to submit a revised version of the manuscript that addresses the points raised during the review process.

In particular, please address the concerns raised by review #2 regarding the validity of your assumptions and treatment of uncertainty. 

We look forward to receiving your revised manuscript.

Kind regards,

Ruth Ashton, Ph.D.

Academic Editor

Journal Requirements:

Additional Editor Comments (if provided):

Reviewers' comments:

Reviewer's Responses to Questions

**Comments to the Author**

1. Does this manuscript meet PLOS Global Public Health’s publication criteria? Is the manuscript technically sound, and do the data support the conclusions? The manuscript must describe methodologically and ethically rigorous research with conclusions that are appropriately drawn based on the data presented.

Reviewer #1: Yes

Reviewer #2: No

2. Has the statistical analysis been performed appropriately and rigorously?

Reviewer #1: Yes

Reviewer #2: I don't know

3. Have the authors made all data underlying the findings in their manuscript fully available (please refer to the Data Availability Statement at the start of the manuscript PDF file)?

Reviewer #1: Yes

Reviewer #2: Yes

4. Is the manuscript presented in an intelligible fashion and written in standard English?

Reviewer #1: Yes

Reviewer #2: Yes

5. Review Comments to the Author

Reviewer #1: Underdosing of antimalarial drugs is clearly a serious problem now and in the future as an accelerant of antimalarial drug resistance. This study is important. It is well written and well explained. I have no major concerns, just some suggestions.

General comments in order:

1. In the abstract, when referring to numbers of cases reported by WHO, please add the word “estimated” in case the reader actually believes the precision of these modelled values.

2. Line 58: “Since the number of malaria cases is rising again, most of the increase occurring in the African region”. Do you mean “since then”?

3. Line 83: perhaps add that the recommended dose could be wrong -many antimalarials have been introduced at doses which were too low.

4. Please define hyperparasitaemia (i.e. >100,000/uL) earlier in the paper.

5. References 22-24 do support the case for the 100,000/uL “increased risk of treatment failure” threshold but it would be very helpful if there could either a figure or a description so that the reader could gauge the relationship between admission parasite density and risk of ssubsequent antimalarial treatment failure. I realise there are many caveats for each of the three referenced studies but, as this is such an important point, the parasitaemia -treatment failure risk does need to be presented in some way in this paper.

6. The introduction provides a strong case for the study. Perhaps add in a sentence to explain or differentiate the risk of underdosing from either low exposure (e.g. pregnancy) or because the standard duration of dosing is insufficient (e.g. hyperparasitaemia).

7. Patients with parasite densities greater than 100,000/uL represent the largest proportion of potentially underdosed patients. The relationship between parasite density and therapeutic response at different levels of malaria transmission intensity and with different ACTs has not been well defined. Please discuss this. WWARN has by far the greatest data sets to identify where we need more information and how confident we can be about this threshold (or more correctly these thresholds). It seems to me that we need more trials specifically in hyperparasitaemic children to determine the threshold for prescribing a longer ACT treatment course?

Please discuss the difficulty in identifying “uncomplicated hyperparasitaemia” if current RDTs are used for diagnosis. Do we need semi-quantitative RDTs to rectify this problem?

Reviewer #2: This is a paper describing estimates of malaria in risk groups that may have higher failure rates following ACT treatment and tries to estimate the number of treatment failures in these risk groups, suggesting that these may contribute to within host emergence of resistance. The assumptions of this analysis are not well supported and are so broad as to make the results somewhat unhelpful. There is no rigorous treatment of uncertainty. Table 1 is presented as if there is no uncertainty. In the real world, the estimates provided here are likely to be completely eclipsed by non-adherence to treatment and suboptimal dosing which is brushed aside with the statement: 100% coverage with ACTs and adherence with ACTs was assumed.

Here is a list of assumptions that are vague, uninformative or incorrect:

- Malaria risk was considered 4 times higher in the rural areas

- Proportion of risk groups within malaria patient populations was assumed to be the same as in the overall country population.

- Recrudescence in risk groups was based on unpublished reviews or assumed to be between 1.2-2.0

o Overweight – assumed to be 1.5

o Pregnant – assumed to be 1.5

- Malaria cases are based on a model and are not validated

- Hyperparasitemia is assumed to be a constant proportion of all cases

- Only selected studies in Table S11 were used?

- 100% coverage with ACTs and adherence with ACTs was assumed.

Note that the methods are two paragraphs, with a 30 page supplement that has to be scoured to really understand what they did.

Suddenly at the end of the results we find this statement –

Three of the five IPD meta-analyses consistently identified that children <5 years of age with uncomplicated malaria but without the above-mentioned risk factors were also at the increased risk of treatment failure when compared to adults (HR 2.68 [95%CI: 1.87-3.85], supplementary material (p21-24) and were associated with an estimated additional number of failure rates of 1.050 - 4.202 million for ACPRs between 98% and 92%.

Which is not part of the methods or objectives, but with an estimated figure that is 2-10 times greater than any of the subgroups in Table 3.

Taken together, these assumptions and methodological flaws render the results to be of limited value.

6. PLOS authors have the option to publish the peer review history of their article (what does this mean?). If published, this will include your full peer review and any attached files.

**Do you want your identity to be public for this peer review?** For information about this choice, including consent withdrawal, please see our Privacy Policy.

Reviewer #1: **Yes: **Nicholas J White

Reviewer #2: No

---

## [Editor Report · Decision Letter 1]

27 Oct 2023

Characterisation of populations at risk of sub-optimal dosing of artemisinin-based combination therapy in Africa

PGPH-D-23-00952R1

Dear Dr. Stepniewska,

We are pleased to inform you that your manuscript 'Characterisation of populations at risk of sub-optimal dosing of artemisinin-based combination therapy in Africa' has been provisionally accepted for publication in PLOS Global Public Health.

Best regards,

Ruth Ashton, Ph.D.

Academic Editor